# Towards Generalizing the Information Theory for Neural Communication [note 1]

**DOI:** 10.3390/e24081086

**Published:** 2022-08-05

**Authors:** János Végh, Ádám József Berki

**Affiliations:** 1Kalimános BT, 4028 Debrecen, Hungary; 2Department of Neurology, Semmelweis University, 1085 Budapest, Hungary; 3János Szentágothai Doctoral School of Neurosciences, Semmelweis University, 1085 Budapest, Hungary

**Keywords:** information theory, neural information, neural computing, neural communication, information content, time-aware computing, neural learning, neural bandwidth, power bandwidth, skewed distributions

## Abstract

Neuroscience extensively uses the information theory to describe neural communication, among others, to calculate the amount of information transferred in neural communication and to attempt the cracking of its coding. There are fierce debates on how information is represented in the brain and during transmission inside the brain. The neural information theory attempts to use the assumptions of electronic communication; despite the experimental evidence that the neural spikes carry information on non-discrete states, they have shallow communication speed, and the spikes’ timing precision matters. Furthermore, in biology, the communication channel is active, which enforces an additional power bandwidth limitation to the neural information transfer. The paper revises the notions needed to describe information transfer in technical and biological communication systems. It argues that biology uses Shannon’s idea outside of its range of validity and introduces an adequate interpretation of information. In addition, the presented time-aware approach to the information theory reveals pieces of evidence for the role of processes (as opposed to states) in neural operations. The generalized information theory describes both kinds of communication, and the classic theory is the particular case of the generalized theory.

## 1. Introduction

Given that we can measure electric signals from outside the brain to inside neurons, it is quite natural to model neural operations with electronic circuits. Since electronic computers exist, parallels are drawn between electronic computers’ operation and the brain’s operation. Principles and notions of electronic communication are used to describe neural communication, despite the many significant differences between technical and biological computing systems. Information content, transferred information, communication bandwidth, etc., are derived for biological systems without checking if the conditions of information theory’s applicability are fulfilled, which, therefore, leads to contradictory conclusions concerning spikes’ information content: “the timing of spikes is important with a precision roughly two orders of magnitude greater than the temporal dynamics of the stimulus” [1] and “the data obtained so far do not suggest a temporal code, in which the placement of spikes relative to each other yields additional information” [2].

It is commonly accepted that “the brain naturally lends itself to be studied with information theory” [3] and neural information science seems to be a well-established theory [4]. However, there is confusion that “the defining mathematical expressions of entropy in statistical thermodynamics and information in the communications field, also called entropy, differing only by a constant factor with the unit ‘J/K’ in thermodynamics and ‘bits’ in the information theory” [5] result in the improper application of the information theory to neural communication [6,7], especially when the constant factor in the formalism with the unit ‘J/K’ (sometimes even ’J/s’) in thermodynamics is changed eventually to ‘bits’ in informatics or vice versa. Although the two notions have the same name entropy, one of them is closely associated with the physical quantities of thermal energy and temperature (and a closely related meaning of disorder in statistical theory), while the other is a mathematical abstraction based on probabilities of messages. The only common aspect seems to be that the concept of entropy is particularly abstract.

In biology, the transferred signal generates electromagnetic and mechanical waves [8], making it harder to distinguish between the entropy of energy needed for transmitting and the entropy of the transmitted information. The confusion increases when using entropy to describe technically implemented electronic information communication, where the laws of physics (the finite speed of light) and the required engineering methods (synchronization and coding/decoding of signals) are not considered; although, they are implicitly considered in calculating channel capacity.

Applying Shannon’s information theory [9] to neuroscience started immediately after the significance of Shannon’s seminal paper was recognized, and different research directions began to use it (for a review, see [10]). However, Shannon warned [11] against the indiscriminate use of the theory and called attention to its valid scope: “the hard core of information theory is, essentially, a branch of mathematics”, and it “is not a trivial matter of translating words to a new domain”. A common fallacy (not only in neuroscience) is to measure an experimental histogram, with arbitrarily chosen bins and transformed variables, and to calculate its entropy, etc., and even to conclude biologically meaningful parameters such as effective connectivity [12,13]. One must also be careful when analyzing time series and concluding causality [14] or synchronization [15] using methods developed for electronic communication. This case is what Shannon warned against [11]: “this is an experimental and not a mathematical fact, and as such must be tested under a wide variety of experimental situations”. In addition to the lack of definition, there are independent arguments (mainly due to the subtleties of the experiments) against using spike-dependent “coding” for cognition-level representation of the brain’s functionality, asking whether “Is coding a relevant metaphor for the brain?” [16]. Biology sees that *the key to biological information transfer is in transferring precise timing information*. Still, the classic information theory cannot consider time as part of the transmitted message. Our research paper does not want to review these contradicting hypotheses and experiments. Instead, we scrutinize the information-related aspects of the neural condenser model, and point out how published measurement data underpin our claims.

First, in Section 2, we compare notions and ideas of communication theory in different fields of application. We discuss the operation of the neural condenser from the viewpoint of information theory in Section 3. Because of their central importance, we discuss the notions of information and entropy in a separate Section 4. We list the pieces of evidence supporting the presented claims in Section 5.

## 2. Communication’s Features

As pointed out in recent papers [17,18,19], in technical computing, the classic computing and communication paradigms behind the reasonable assumptions and formulas (including Shannon’s classic idea on information transfer) prove more and more to be a rough approximation. Good information theory textbooks (such as [20]; we are using their terminology and notations) exclusively discuss the mathematical theory and formulas describing entropy, both for discrete and analog (i.e., density functions) quantities. (Using quasi-analog variables do not change the discussion significantly). We note that a section-wise uniform distribution density function integrated over a closed interval is also perfect to describe the entropy of communication.

In this section, we compare how different fields interpret some notions of information theory and which neural features make their use doubtful or erroneous. We aim to show how one shall extend Shannon’s basic notions before using the information theory to neural communication. We discuss the aspects of the neuronal condenser in more detail.

### 2.1. Communication Model

The technical model (the mathematical model was inspired by the technical implementation) of communication theory uses well-defined components, model, and functionality; for details, see [9]. When we speak about “communication”, we refer to Shannon’s five-element communication model [9] and his definition: “The fundamental problem of communication is that of reproducing at one point either exactly or approximately a message selected at another point”. We do not repeat the discussion available in [9], only the deviations due to the transferred information or the communication medium.

In neuroscience, “the interpretational gamut implies hundreds, perhaps thousands, of different possible neuronal versions of Shannon’s general communication system” [7], and so, that of the entropy. It is not clear whether neuroscience commonly uses one of the above models and the notion ‘*entropy*’ or another—neuroscience-specific—one(s). The less standard models of communication, probably the effect of the models used in neuroscience, started to appear in connection with artificial neural networks, see [21].

### 2.2. Mapping Sender(s) to Receiver

The electronic communication model assumes one sender only, while several senders (the presynaptic neurons) participate in neural communication. This difference alone makes the applicability of the classic information theory to neural operations questionable. If we consider the spike as a message in Shannon’s sense, we need to modify this assumption to consider this multiplicity. In the mathematical model, we need to generalize the 1:1 mapping of senders and receivers to M:1 mapping. Given that the information carrier is a voltage level in the technical model, we must interpret how multiple voltages could be connected to a single electronic component. In our model, the neuronal condenser naturally accepts many input messages (currents) and provides a single output message.

### 2.3. Reproducing a Message

In the mathematical theory, there is no problem with Shannons’s requirement to reproduce a message in the receiver. In electronic communication, the construction of the system enables us to guarantee a similar frequency of the internal oscillators in the sender and the receiver. Quite similarly, in biology, the communicating neurons are anatomically similar so they can reproduce the signal, but the biological mode of operation restricts that activity. In our model, the “control unit” (depending on the actual potential of the membrane) can close the ion channels in the synapses. To avoid admitting that information can be destroyed, one needs to introduce the term, closed (isolated) system. If the information produced by the source system cannot get through the communication channel, it will go to the outside world. That is, it will not be lost, but the circle of actors in question must be extended to the environment.

Consequently, the input message (the presynaptic neuron’s output spike) can be reproduced neither at the postsynaptic side of the synapses nor at the neuronal membrane. In the latter place, the input signals can also interfere. Whether the receiver neuron can fulfill Shannon’s requirement to reproduce the message is questionable.

### 2.4. Information Content

One of the basic assumptions of mathematical theory is linearity. Shannon’s “logarithmic measure” assumes that “two punched cards should have twice the capacity of one for information storage” [9]. In neural communication, “two spikes close together in time carry far more than twice the information carried by a single spike” [22]. That is, if we consider the spikes as information storage units, for neural communication, we must not use Shannon’s suggested “logarithmic measure” for information. Fortunately, a message can also be “a function of time and other variables” [9]; although, Shannon did not discuss this case. The present paper suggests extending Shannon’s theory in the direction of including spike-related timing in the messages sent by the neurons. We apply the time-aware computing paradigm [17] to our specialized neural condenser model.

### 2.5. Signal Propagation Speed

One of the possible roots of the experienced incoherence in information theories applied to different fields is *the speed of communication*. In mathematics, there is no signal propagation speed. There are logical successions only, and the signal appears promptly in the receiver. Introducing a finite transmission speed also implies introducing temporal alignment and transmission delay. In electronic communication, the signals propagate at a speed proportional to the speed of light. The high speed of electromagnetic waves and the usual distances between components make the “instant interaction” a more or less acceptable approximation. However, the effect of using time-unaware description is more and more visible, from the very high thermal dissipation of processors [23] to the performance gain rooflines of supercomputers [24] and performance limitations of artificial neural networks [25].

In neural communication, the transmission speed is about a few million times smaller than that of the electronic transmission, so the transmission time is not negligible, even though it can dominate communication. Von Neumann told [26] that using his suggested time-unaware description to neural communication was “unsound”. The “spatiotemporal behavior” is noticed, but its common description misses the point that signals’ time and position are connected through their conduction velocity. Considering transmission delay is becoming important in both biological [27] and large-scale technical computing [24,25].

### 2.6. Bandwidth and Noise

The reproducibility requirement means that one must produce a signal at both the sender and receiver sides; furthermore, the sampling theorem claims that the signal’s frequency must match the frequency of the signal propagating through the channel. In a well-designed system, the transfer channel must be able to deliver the received density. The smallest one of the named density limits defines the density that the system can work with: its bandwidth.

To define a “bandwidth”, we need to consider two signaling features: how quickly we can generate a message (signal frequency) and how quickly the message can propagate in the respective medium (signal propagation speed). These features depend on material characteristics and science laws. Some (time-dependent) material characteristics, usually an amplitude, describe the signal. The sender’s ability to produce an information density is limited. The receiver has to reproduce the signal, so its ability to receive an information density is also limited.

Given that a medium cannot have two different material characteristics at a given time at a given place, a signal must entirely pass at a given position before the next signal can arrive at the same point. We can define the theoretical channel capacity as the signal frequency in which the signal periods follow consecutively, without overlapping and creating gaps between them. That frequency represents the channel’s theoretically possible maximum information density in units [number of signals/time].

In mathematics, neither propagation speed nor wavelength has a limitation. They are just arbitrary functions without having any connection with each other. Because of this, in mathematics, only an arbitrary bandwidth can be interpreted. Shannon, in his famous work, introduced the notion of “the number of allowed signals of duration T” without deriving it; he only showed how one could handle the bandwidth mathematically if it is known from some external source, such as the ones provided by Nyquist [28] and Hartley [29]. Shannon’s work implicitly refers to electronic communication’s speed and frequency.

In electronic communication, “the signal is perturbed by noise during transmission or at one or the other of the terminals” (an unusual characteristic of Shannon’s idea is the way of considering noise in the transfer channel: it introduces a time-averaged contribution to the information transmitted through the channel instantly) [9]. For the sake of simplicity, we consider transferring binary information in a system controlled by a central clock over a finite-bandwidth continuous-time channel. We tacitly assume that the speed of the physical carrier cannot vary. The generosity of the requirement for the sent and received messages, “although an exact match is not in general possible; we can approximate it as closely as desired” [9], enabled us to reproduce the signal also in the presence of a reasonable amount of noise. The presence of noise reduces the bandwidth, as discussed in the literature [20].

Given that, in biology, both the maximum firing rate and the spike length are defined anatomically, we can calculate the limiting bandwidth of the transfer channel (axon) in the same way as in the case of technical transfer. The partner neurons have anatomically similar RC constant, providing a sufficiently similar time-base in the sender and receiver neurons at the two ends of the communication channel. For safe communication, both the sender and the receiver and the axon between them shall be able to handle the highest possible frequency (information density). The evolution forced the partners to tune their parameters correspondingly: information loss has fatal consequences. Presumably, this natural neural bandwidth is only slightly above the maximum possible information density the transfer needs.

### 2.7. Transmission Channel

From the anatomical viewpoint, the axon belongs to the presynaptic neuron; while in the communication model, the axon as a communication channel is an independent entity with its attributes. Shannon mentions only *passive* channels: “the channel is merely the medium used to transmit the signal from transmitter to receiver” [9] and that “the source as seen from the channel through the transducer should have the same statistical structure as the source which maximizes the entropy in the channel” [9]. In biology, the above bandwidth limits the information transfer through the channel essentially in the same way as in the case of electronic communication, except that the conduction velocity defines the signal propagation speed, and the spike’s temporal length defines the signal’s wavelength. Given that in the neural communication process—in contrast with the technical implementation—the axon actively participates in transmitting spikes, a *local energy supply* along the transmission path is also needed. The energy derives from a local resource (ATP) [8,30], which must be available to forward the signal, and that resource needs a regeneration time. This requirement forms a “power bandwidth”: the message which requires more energy supply than locally available, will not be able to get through. To establish our neural communication model, we need to introduce an additional attribute, “transmitting ability”, for the transmission channel. Either the communication bandwidth or the power bandwidth can limit the information density of neural communication. If we attempt to transfer information with a density exceeding the communication bandwidth, the produced information cannot reach the target system entirely. The case is similar to the ‘roofline’ effect experienced in technical computation [31]. For details, see Section 5.3.

We can safely assume that under natural conditions, the density of energy supply along the axon is just slightly above the value required to transfer the maximum information density the presynaptic neuron can produce. These parameters are properly concerted under natural conditions (in healthy neural systems), and no information loss occurs. However, if the parameters are changed (because of illness or invasion), the transmission operation may result in information loss, and the neural network shows the symptoms of abnormal function.

### 2.8. Synchronization

Shannon requires to reproduce the message by the receiver. The vital fact that *reproducing occurs at a later time* remains entirely out of scope in mathematics; that is, it assumes that the transfer happens instantly. Given that the receiver needs to reproduce the message, technical and biological implementations must provide a mechanism to synchronize [32] the receiver to the sender; among others, to detect causality [14,15]. Since initially the sender and the receiver run independently, one cannot avoid the initial synchronization; therefore, the partners must send and receive some service information. Synchronization must occur at least once in all communication steps (in every signal train). The electronics communication tacitly introduces a synchrony signal: a “ring signal” (in different technical implementations) must define the beginning of the message’s arrival. Quite similarly to the role of temporal alignment in the general computing model [17], the synchronization may be explicit (in the form of a separated physical signal) or implicit. The lack of synchronization in the mathematical theory at least deserves special attention when applying it to electronic or biological communication.

The synchrony signal is a piece of information that must be transmitted. Still, it does not belong to the *payload information* in the classic sense. Correspondingly, a transferred message comprises non-payload and payload information and has a joint information density limit. The payload information density limit is less than the theoretical maximum. The efficiency of transfer (i.e., the payload bandwidth) also depends on the ratio of payload to total transferred bits. The sampling theorem does not differentiate the bits by their role. One can consider the above assumptions: the system needs some time to transfer payload information, some time for service information, and some extra time to transfer technical noise. Because of this, the presence of noise and the need for synchronization naturally requires a portion of the total bandwidth, reducing the usable payload bandwidth.

The crucial point is that *for reproducibility, the sender and the receiver shall be synchronized*. In technical transfer, a rising edge (such as a clock signal) or a specific pattern signals the information’s arrival, and the phase of information decoding gets fixed upon the message’s arrival. The actual time, relative to a spike’s beginning, is the same in both the sender and receiver systems. The transfer speed and the distance between the systems also do not have a role; at least, if one sender and one receiver are assumed. Given that usually many senders contribute to charge collection, the time of arrival matters. Synchronization is needed to define the ’bit time’ and enable the communication to be independent of the distance of the communicating systems and the propagation speed between them. Shannon’s original idea assumed instant interaction between the sender and the receiver circuits. In our interpretation, after synchronization, *Shannon’s theory remains valid for any transfer speed and distance, provided that the two systems are appropriately synchronized*.

Unlike in technical transmission, because of its 10M times lower transmission speed, in neural transfer, it is explicitly considered that sending and receiving a signal occurs at different times. The sender initiates a transfer when it needs to, and the receiver starts receiving the signal when it arrives at its destination. Because of the anatomical similarity, their timing characteristics, RC, of the neurons are similar, but their initial phase must be set. The arrival of a spike provides an autonomous auto-synchronization signal (part of the non-payload information transfer) in the receiver; it sets the signal’s phase in the receiver to the phase in the sender. The ‘end of computing’ signal is triggered either by the normal charge-up process of the membrane or by the additional mechanism of exceeding the synaptic current threshold [33]. The latter (hardly observable) mechanism assumes a transient irregularity in the firing rate—a mechanism not yet reported.

The mixed electrical/chemical charge transfer mechanism delivers a charge in the spike towards the membrane, but it cannot enter because the input conductance is zero. However, the stalled ions create a potential that opens the potential-controlled ion channel (this is formulated as “Synaptic inputs effectively open ‘holes’ in the membrane” [34]), then the charge in the spike can contribute to the membrane’s potential. That is, *the spike itself comprises its synchronization information*. A new ‘computation’ starts when the first spike arrives. The arrival of the spike is also the ‘ring signal’. In this sense, transferring synchronization information does not need extra bandwidth.

### 2.9. Information Distortion

Biological information has amplitude and time attributes. As their operation discussed, neurons actively participate in processing neural spikes. The amplitude of the ionic current and spikes’ arrival times affect the membrane’s charging process. Furthermore, the charging behavior of the target neuron also contributes to the neural noise. In the neural condenser model frame, neural noise leads to deviations in ‘bit time’ values and total transfer time. Given that all communication features are defined anatomically, any change in ion/ATP concentration, neural distance, conduction velocity, anatomical environment (including the invasive ones), etc., can lead to a distortion in neural communication.

## 3. How the Neuron in Our Model Works

The biological signal transfer uses a mixed chemical/electrical mechanism for the signal transfer (meaning about 10M times slower transmission speed and non-Ohmic mechanism) and uses a non-discrete signal form; the signal’s amplitude is not only carrying the information, and many senders can send a message to the receiver. Notice also, regarding the new element compared to technical transfer, the communication channel actively participates in the transmission: the axon’s ion channels pump ions in and out, needing a distributed local power supply and introducing an independent *power bandwidth limitation*. Although it is also true that “the emergence of life cannot be predicted by the laws of physics” [5] (unlike the creation of technical systems), models in neuroscience (as reviewed in [35]) almost entirely leave the mentioned aspects out of scope.

### 3.1. Neuron Model

Our model essentially uses specialized neuronal condensers connected with finite-speed transfer lines which need a local energy supply; we discuss their operational characteristics in the subsections below. From a communication point of view, we consider neurons comprising an input section, a memory, and an output section, running under the control of an internal control unit. The presynaptic neuron’s “output section” corresponds to the sender+encoder unit, the axon corresponds to the transfer channel, and the neuron’s “input section” (its synapses and the membrane) correspond to the receiver+decoder unit. The neuronal membrane can have an excess potential (compared to its resting potential) in the relative refractory period, so it can represent a memory (with time-dependent analog content). The control unit (represented mainly by voltage-controlled ion channels) regulates the in-flow through the synapses and triggers the output signal (an action potential (AP)).

### 3.2. Neural Memory

Given that the internal operation of the neuron membrane needs time (the neuron membrane is not isopotential [36,37]), the result of a neural computation depends not only on its inputs but also on the neuron’s internal state (a spike arriving during the relative refractory period finds a membrane potential above the resting level, leading to spike generation at an earlier time). One can easily interpret the effect using the temporal behavior [38], seen experimentally as nonlinear summing of synaptic inputs [34]: the changed time of charge arrival (due to changes in the presynaptic spiking time, conduction velocity, and synapse’s joining location). This time role suggests that *biological computation results in changes in neurons’ temporal behavior, instead of spike signal amplitude,* as expected from the parallel with technical systems.

The validity of using a memoryless (Markov) process to describe neuron’s computing depends on its firing rate: as the synaptic input rate increases, the neuron membrane’s potential at the beginning of the next computing cycle tends to deviate from the resting potential; the *neuron has a memory.* (The effect is in resemblance with that of the memristor [39,40]. The question remains open in both cases: how much can the effect be attributed to material/biological features and the lack of temporal behavior in their description?) [27]. The presence of memory is noticed (for example, “deviations from the equilibrium [membrane potential]” is mentioned as “a form of intracellular memory” [30]), but, because it is hidden (it is an internal feature of the neuron), it “cannot be injected directly into a neuron in an in vitro setup or into a model neuron” [41].

When attempting to tackle the problem of quantifying information for time-dependent signals, the first idea was to use signals with multiple states (notice the lack of temporal behavior: there are no processes, only states). Given that the result of a neuronal operation depends on both the neuron’s inputs and its internal memory, both of which are time-dependent, simply introducing vector representation for the multiple input and memory states leads to a dimensional explosion [2], blocking the practical usability of the method. This situation troubles to measure and interpret dynamic characteristics of a neural network, such as effective connectivity [13].

In practice, neuroscience focuses mainly on the average response of cells and their neural noise. However, neural communication’s precise, timely internal nature is poorly understood and generally omitted. The approach can be slightly enhanced if one uses statistical assumptions on the distribution of amplitudes, but such a method blocks concluding the effect of individual stimulations. Further limitations are also discussed in [2]. If one assumes an absolute time (as opposed to spatiotemporal distance) in a biological system, one must introduce such a complementary method. Considering that neurons’ temporal behavior may have a crucial role in describing the mechanisms of information transmission, the question remains open: whether the information is coded in selecting one of the multiple states.

### 3.3. Neural Bit Time

When the neural computation starts, the current from its synaptic inputs starts to charge the neuron’s membrane. The neuronal condenser’s control unit can select how much from the (time-dependent) input density functions (currents), it integrates into the actual measurement. When the membrane reaches its threshold potential, the receiver neuron closes its input channels and starts to prepare [37] its spike. *The result of the computation, a single output spike, may be issued when neither of the input spikes was entirely delivered*. The rest of the received synaptic charge is lost (the neuron pumps the ions out to the extracellular space), i.e., the charge (integrated signal amplitude) will not be conserved if the extracellular space is not included in the closed system. In this interpretation, the information (represented by charge) sent by the presynaptic neurons may be lost, and a piece of different information is created independently by the neuron. In the conventional interpretation, the only information from the neural computing process is that the membrane’s potential exceeded its threshold value: precisely, one bit. However, the vital information, *when* the neuron generates a spike, depends on the actual state of the received presynaptic inputs and the neural memory. The neurophysiological measurements correctly conclude that evidence.

Given that the circuit will spontaneously fire in biological systems, some noise is present, i.e., the “bit time” has an upper bound. Given that the neuronal condenser needs time to be charged and discharged, the bit time has a lower bound. If the input current persists (in a stationary state), the firing frequency will depend on the input current and have an upper frequency limit. If the condenser has multiple (time-dependent) input currents (in a transient state), the output frequency will reflect *the change in all of its inputs* (i.e., the effect of the network). An observer measuring the firing rate of our condenser periodically measures the effect of the processed probability density (integrated charge density) functions that the condenser receives. Measuring ‘firing rate’ assumes *states* but no *transitions* between them. The spikes are essentially sampling signals. This is no way to make sampling measurements with time resolution finer than the spike’s length.

The transfer speed (conduction velocity) is not constant in the long run. In long-term learning, biology may myelinate the presynaptic neuron’s axon at its end near the synapse [42]. The absolute length of our ‘neuronal bit time’ may vary at individual neurons. Our definition, however, is not sensitive to such changes: myelination results in slightly higher transfer speed (and so: slightly shorter ‘bit time’), but our conclusions remain valid. Notice that the bit time is defined at neuron level and it is not necessarily related to bit times in its presynaptic neurons, so we cannot interpret a neural network-wide bit time. Our interpretation is in line with the conclusion that “the neural code… may be significantly context dependent” [1].

### 3.4. Firing Rate

The momentary value of the time of the total operating cycle of the neuron is given as
(1)$$T_{Computing}=T_{Triggering}+T_{Charging}+T_{Refractory}$$
where $T_{Triggering}$ is the time which neuron needs to collect the potential contributions from its synapses to reach the threshold, $T_{Charging}$ is the time (according to [37], it is about 200 μs) needed to charge the membrane to its maximum potential, $T_{Refractory}$ (absolute and relative, in the range of 1–3 ms) is needed to reset the neuron’s initial state. Here, we assumed to start the time measurement at the beginning of the computing process. However, usually some idle time $T_{Idle}$ (no neural input) also follows between the computing operations, making $T_{Computing}$ longer. That is, the operating frequency (its firing rate, the reciprocal of $T_{Computing}$ of the cyclic operation) is
(2)$$F=\frac{1}{\underset{Memory+Network}{\underset{︸}{T_{Triggering}}}+\underset{Neural\;anatomy}{\underset{︸}{T_{Charging}+T_{Refractory}}}+\underset{Network+Neuron}{\underset{︸}{T_{Idle}}}}$$

Given that $T_{Charging}$ and $T_{Refractory}$ are defined anatomically, the only way to adjust neurons’ firing rate is to change $T_{Triggering}$ and $T_{Idle}$ times. The firing rate has two degrees of freedom (in other words, the information is shared between two channels). $T_{Idle}$ is relevant only in producing *spike trains* (periodic operation): how long the ‘pause’ between computations is enabled. The absolute refractory time of the $T_{Triggering}$ is relevant in internal neuronal processes: it depends on how quickly the neuron’s membrane reaches its activation threshold potential after receiving its first synaptic input. Notice that measuring spike-related data forces a sampling-like measuring method on the experimenter.

Finally, the neuronal memory depends on the network’s former activity; we can say that the two degrees of freedom describe the networks’ activity from two different viewpoints. A complex convolution of the past (a weighted integral of the respective network activity in the past relative refractory period) and present network activity defines the momentary firing frequency of the neuron. We cannot understand a neuron’s dynamic operation without considering its network: the neuron is an unusual simple processor which takes its inputs from its time-dependent environment. The opposite claim is also valid: we cannot describe a neural network without accounting for the temporal behavior of its neurons. Our analysis is in line with the claims of Brette [16], that the “neural code depends on experimental details that are not carried by the coding variable” and “spikes are timed actions that mediate coupling in a distributed dynamical system”.

The (anatomically defined) maximum operating frequency is
(3)$$F_{Max}=\frac{1}{T_{Charging}+T_{Refractory}}$$
which is *a feature of the individual neuron* only, while *F* characterizes it in *its specific network environment*. Contributions $T_{Triggering}$ and $T_{Idle}$ are convoluted in the measured firing rate: their effects are mashed when measuring *only* spiking rate represented by *two* probability variables, average and variance. When the effect of $T_{Triggering}$ and $T_{Idle}$ are measured as one single (reciprocal) random variable *F*, there is one attempt to describe *two* random distributions, with the average and variance of *one* (reciprocal) distribution *F*. In other words, one attempt to describe four parameters with two parameters only. However good the description could be depends on the experimental conditions. If, for example, $T_{Idle}$ can be experimental or, during the evaluation, omitted (for example, see Section 5.1), we can adequately describe the resulting distribution by one probability variable.

### 3.5. Learning

A neural condenser can adjust its time $T_{Triggering}$ either by collecting more charge to its membrane from its synapses or collecting the same charge in a shorter time or by combining these two methods. (Given that several synaptic inputs contribute to the inter-spike timing, the neural network can work much faster than its component neurons (“roughly two orders of magnitude greater than the temporal dynamics of the stimulus” [1]); several neurons can cooperate.) Biology provides mechanisms for both ways of adjusting the triggering time. The two methods differ in their implementation speeds and operating costs, enabling the biological systems to find a proper balance between the two.

One mechanism is to increase the synaptic weight $W_{i}$ corresponding to its *i*-th synapse. It can be carried out by changing the shape of the conductance function $gsyn_{i}\left(t\right)$ [43]: a broader function (providing more transmitters) has such an effect. The ions in the spike anyhow reach the target at its presynaptic point. Increasing neurotransmitters’ local gradient means delivering more charge to the (postsynaptic) membrane, causing a more significant increase in the membrane’s potential. The neuron must pump out the ’unused’ (not transmitted to its membrane) ions, which is a rather energy-consuming action. However, tuning the conductance functions gsyn(t) [43] is fast: increasing transmitters’ concentration can be implemented in a few *msec* periods. As observed, “elementary channel properties can be rapidly modified by synaptic activity” [44].

Another mechanism is to decrease the time needed to transfer a spike from the presynaptic neuron to the target neuron: biology can change the corresponding axon’s thickness. (The mechanical waves [8] can slightly change the length of the axon. This effect can also contribute to transfer time and presumably increases its standard deviation). Given that the transfer speed (conduction velocity) on thicker axons gets higher, the spike arrives in a shorter time and, in this way, contributes to the membrane’s potential at an earlier time (i.e., the membrane reaches its threshold potential earlier). This mechanism is less expensive in energy consumption but needs a significantly longer time to implement. The aspect that changes in conduction velocity “could have profound effects on neuronal network function in terms of spike-time arrival, oscillation frequency, oscillator coupling, and propagation of brain waves” [45] and that “node of Ranvier length [can act] as a potential regulator of myelinated axon conduction speed” [42,46] have been noticed. The myelinization begins at the axon’s end nearer to the postsynaptic neuron, and the postsynaptic neuron triggers this change. On the one side, it is a kind of feedback for communication; it acts on the postsynaptic neuron’s operation without affecting the presynaptic neuron but changing the network operation. On the other side, from an anatomical viewpoint, the axon belongs to the presynaptic neuron; however, from an information theoretical viewpoint, the postsynaptic neuron changes the presynaptic axon’s features.

Given that both mechanisms result in shorter $T_{Triggering}$ times, they cause the same effect: the computing time (or, in other words, the firing rate) changes, and one cannot tell which mechanism caused it. This equivalence is why nature can combine them: a neuron can increase its firing rate quickly (as a trial) and (on success, if the learned condition is durable) it may decide to reimplement (remodel [47]) its knowledge in a less expensive way. It makes the corresponding axon thicker. After that, the needed weight $W_{i}$, which was increased for the short-term learning (experimental) period, can be decreased to conserve the learned long-term (stable) knowledge while reducing the energy consumption. The effect (the learned knowledge) remains the same. Still, the role of time in storing information and learning has not yet been discussed.

Combining these mechanisms provides a way of learning: the spike which contributes its charge at an earlier time or its current gets higher as the increased conduction speed has more influence on the membrane’s potential. That is, it contributes more “information” to the bit (spike) prepared by the neuron, making its spiking time shorter. Hebb’s observation describes that the receiver neuron increases the proportion of the ‘good’ synaptic inputs. Notice that the proportion of all other synaptic inputs decreases; the mechanism provides a ‘biological renormalization’, too. Also, the mechanism explains that some pre-wired information contribution exists and that neurons can learn to evaluate their information sources. Furthermore, these two types of information contributions are of equal rank [48].

This double mechanism is expressed as “we should not seek a special organ for ‘information storage’—it is stored, as it should be, in every circuit”. Mainly because “information stored directly at a synapse can be retrieved directly”. Besides, our analysis adds the discovery that remained hidden: the information is stored through handling the time or, in other words, adjusting temporal processing. Biology takes advantage of the low speed of information propagation; one more piece of evidence for the neural design principle is as follows [49]: “Send only information that is needed, and send it as slowly as possible”.

We conclude that *the modulated signal’s delivery time changes the received message compared to the sent message* (in Shannon’s notion: introduces non-zero entropy contribution). The axon is not a passive transfer channel: it actively changes the message content (at least the used portion of it). Neural learning has serious implications for the neural information science, too. Shannon’s expectation that “the source as seen from the channel through the transducer should have the same statistical structure as the source which maximizes the entropy in the channel” [9] is not fulfilled.

Given that the computation starts on receiving the first input spike, if the beneficial transfer speed changes, it may put another synaptic input into the position of providing its spike before the previous first synaptic input. The output spike will be delivered to the next neuron at a different time, but the auto-synchronization mechanism will function appropriately in the next neuron. The communication mechanism is transparent for learning because of the synchronization.

## 4. Information and Entropy

### 4.1. Representing Information

In mathematics, the information to be transferred is knowledge on the actual state (out of several possible ones; using a continuous probability variable does not change the landscape significantly) from one entity to the other, without assuming explicitly any transfer channel and without the need to code/decode that information. Because of assuming instant interaction, temporal information (at what time the knowledge was valid) is not included in the message. The reason why we use the logarithm function, according to Shannon, is not absolute: “logarithmic measure is more convenient… it is practically more useful… it is nearer to our intuitive feeling… it is mathematically more suitable”; furthermore, “any monotonic function of this number can be regarded as a measure of the information” [9].

The electronic information communication theory tacitly assumes that prompt discrete (binary) transfer takes place and that the *signal amplitude* corresponds to the value of the discrete random variable in the mathematical theory. The spatiotemporal behavior is noticed: the messages from a larger distance arrive at a later time. From the information theory’s viewpoint, the received message comprises a digital signal plus a timing portion. The timing portion is not modulated (according to Shannon, it has zero entropy), so no information is lost if the timing is eliminated. Correspondingly, clock domains and distribution trees are used to suppress the timing portion of the message. The tolerable temporal inaccuracy of signaling (“clock skew”) is approximately 5% [50]. That is, in technical systems, the timing portion of information belongs to the message but it is neglected (or compensated by tunable delay lines [51] in biology-mimicking chips), according to the time-unaware paradigm. Omitting the temporal behavior limits payload performance, among others, in supercomputing [24] and artificial networks [25].

In biological communication, easured evidence shows that timing precision matters. For example, timing enables the extraction of behavioral information: “the coincidence detection property of a spiking-based feed-forward neural network enables mirror symmetry” [52]. Experiences that “synergy between spike trains can be measured” and “two spikes close together in time carry far more than twice the information carried by a single spike” [22] suggest that *single spikes and their relative timing share the transmitted information.* (Combining these measured quantities into a monotonic merit function to describe neural information, as expected by Shannon, would be a real challenge.) Based on the ‘fixed bit time’ method, one cannot interpret the evidence that several bits can be transferred in a single spike and even more in spike trains. Given that those contradictory conclusions all use the neural information theory, based on the analogy with technical information transfer, it is worth scrutinizing if *the false parallel itself may lead to different conclusions under different conditions*.

In our view, a real-world Shannon’s message comprises both time and signal components and both components can carry information. However, it is hard to combine those two pieces of information into one single monotonic merit function as Shannon required, so one needs to use some approximation. In technical communication, the messages’ delivery times depend only on the cable length between the partners, so that the sender cannot modulate it (cannot use it to transfer information independently). Consequently, *in the technical communication, it is a good approximation that only the signal component carries information*. The difference in the messages’ arrival times is not only non-informative for the receiver, but it also disturbs decoding the payload signal, so the technical implementation can suppress the time component without losing information. Furthermore, if the information sent is about technical states (such as high/low level of a signal), measuring the information in ‘bits’ is appropriate.

The mathematical theory was inspired by the technical communication and was targeted to establish its mathematical description; so, in the mathematical handling, the time component is simply omitted.

In the biological communication, the propagation time is modulated by the partners, while the signal part is defined anatomically (in Shannon’s notion: introduces non-zero entropy contribution). Consequently, *in the biological communication, it is a good approximation that only the time component carries information*, and it does so in the form of delivering a temporal distribution. Whether the information transmitted in this way can be measured in ‘bits’ (for example, as suggested by Equations (Equation 4) and (Equation 5) is still an open question and must be verified experimentally. Our estimation is that we need to introduce the notion “neural entropy” (i.e., one more kind of entropy) for the neural information transfer. How this new notion can correspond to the “electronic entropy”, must be scrutinized. Furthermore, deriving information and entropy for biological communication from the number of spikes (or derivatives missing the time component) has as much relevance as would be so in technical communication from the length of cables connecting the parties.

Re-interpreting neural information will probably cool down the fierce debates concerning the nature of coding neural information (for discussion, references, coding types and their evaluation see Section 3.8 in [4,16]. Different measurements seem to underpin other theories, mainly because “the neural code depends on experimental details that are not carried by the coding variable” and “spikes are timed actions that mediate coupling in a distributed dynamical system”, as analyzed in [16].

### 4.2. Measuring Information

In a mathematical sense, we take a value of a probability variable in the sender system and evaluate the received value in the receiver system; no measurement is included. In technical and neural systems, we need to define a measurement method. The fundamental issue when measuring *probability* at a point, is that we calculate the integral of the corresponding *probability density* over a region containing the point. To provide a possibly accurate momentary value, on the one hand, one must choose the size of the region as small as possible. On the other hand, the technical limitations define the minimum time required by the measurement procedure, which allows us to tell the value of the requested level with sufficient accuracy. If we measure the information in bits, this time is the bit-time.

According to the sampling theorem, in practice, the bit time is determined in the opposite way: the maximum bandwidth the system can transfer defines the frequency of the transfer and that frequency defines the bit-time; the signal parameters shall be adjusted to produce a measurement value with sufficient accuracy. The classic measurement method to find out the received message is taking ‘samples’ from it with the frequency calculated using the method above; that is at a temporal distance defined by the bit-time. In the electronic transfer, the speed of electromagnetic pulses (with their ≈2/3 light speed) is implicitly assumed, which would be obviously wrong for the neural transfer having its conduction velocity in the speed range of m/s. In the electronic transfer, the frequency is defined by material characteristics and can be varied; while in the neural transfer, the maximum operating frequency is defined by the temporal length of the spikes.

When using this bit-time definition, we tacitly assume that *the signal amplitude carries the information* and that *the two systems run synchronously*. Furthermore, we need to define the *bit-width* (having the dimension of time) of communication on both sides. Using this method, we consider that *measuring the signal’s amplitude* defines the transferred bit, and the internal oscillator provides the signal’s ‘end of bit time’.

Alternatively, we could integrate the received current for the duration of the bit width. That is, we use a differential entropy: a time (or voltage) measurement (instead of a constant frequency oscillator) provides the ‘end of bit time’ signal. We shall formulate the “bit time” differently. A bit is delivered when the expected charge is collected (event-driven instead of time-driven); the comparator connected to the condenser signals, exceeding the predefined voltage threshold value, and the receiver circuit “fires”. If the signal’s current level is constant, this method delivers the same bit time (and noise) as the classic one. The amount of transferred information (including the noise), represented by the amount of charge, remains the same with both definitions: one bit. However, if the received current is not constant, the “bit time” can be different for the individual bits: this method is a generalization of the sampling method.

Although the charge integration essentially corresponds to calculating the entropy from a distribution function (see the textbook [20]), it differs from it in two essential points. First of all, it is not a probability density in a mathematical sense and it does not deliver state information. As we discussed above, the biological transfer works differently. Second, in our case the “right support set” over which we integrate, is defined by the result of the integration: when the threshold reaches the “control unit”, it terminates the charge integration. The mathematical formalism shall be modified correspondingly.

### 4.3. Conservation Laws

In the mathematical communication, a message created in one system (maybe some noise added) entirely appears in the other system. Despite that the communication units have well-defined functionality, mathematics assumes reversible processes, instant interaction, and direct information exchange without an explicit role for the communication channel and an arbitrary limiting rate on information exchange. This handling enables to introduce mutual information, although the model only enables a system to be either the sender or receiver in a single logical step.

In technical and biological implementations, only one-way transfer channels exist. Scrutinizing the details of processes reveals that in the case of any implementation, we need to consider what mechanisms the sender, receiver, and communication channel apply and how the terms in their technical and biological implementation can correspond to the notions of the mathematical abstraction. Special consideration is devoted to the law of conservation of information (contrasted to the conservation of entropy, which in one meaning, contradicts the second law of thermodynamics, and in the other meaning, no law of conservation exists) through considering the conservation law of the physical quantity by which the information is represented.

As presented above, the law of conservation of information is fulfilled (provided that the transferred charge represents it and the timing conditions are appropriately set): the charge corresponding to one bit is transferred from one system to another. However, a neural condenser may have several synaptic inputs with randomly timed currents. The condenser will fire when the integrated charge exceeds the value corresponding to one bit irrespectively, and the‘bits’ sent by the presynaptic neurons are not yet entirely delivered. (This behavior suggests sending the overwhelming majority of the information in the front portion of the spike.) Given that the information is (mainly) represented by time, there are no issues with a conservation law.

### 4.4. Location of Information

Our idea also gives way to interpret where the information is located while it is transferred from one system to another. In mathematics, there is no such notion: the information immediately appears in the other system. In a technical implementation, if we integrate the density function at a given position of the space over a period including the corresponding bit time, we receive the bit at that position. We cannot locate the information more precisely; measuring its time and place simultaneously is in parallel with the uncertainty relation known in modern physics.

In neurophysiology, it is evident from the very beginning that the neural signal is extended (as opposed to point-like), and it can be located along its path as the AP travels from one neuron to another. The value of an AP can be recorded at any point along its path, and the AP is seen at different times at different locations: the behavior of signal propagation is spatiotemporal.

A similar behavior should be seen when communicating with electronic signals: the spatiotemporal behavior also manifests in those systems. However, because of the 10M times higher signal speed, we should have a time resolution of dozens of picoseconds instead of milliseconds. The wiring length between communicating components ranges from the typical several centimeters to, in supercomputers, dozens of meters. When using 1 GHz transmission frequency, the bit time is about 1 nsec, and we can localize the transferred information in a space region of several centimeters. The distance of communicating components and their accuracy of information localization are in the same range: nsec times. However, in the case of Ethernet(-like) networks, the time distance can be in the range of msec times, which must be considered when running time-aware simulations.

The electronic network models do not consider the mentioned spatiotemporal behavior: the messages arrive to the peer nodes at the same time, in the next time slot. Not using spatiotemporal behavior means assuming an infinitely high signal propagation speed, an unrealistic assumption for a physical network. The disadvantageuos effect of neglecting the spatiotemporal behavior effect is seen primarily in vast supercomputers [24]: the non-spatiotemporal description led to entirely irrealistic efficency and performance gain; the more unrealistic, the larger the supercomputer is.

When scaling down transmission frequency to 100 Hz for biological systems and transmission speed to 10 mm/ms, furthermore, estimating ’bit time’ from a 10 ms spike length, we see that we can localize the information in biological systems with the same few cm accuracy. Given that the neural information is shared between spikes and spike trains, the “size” of neural information can be relatively much larger than that of electronic information.

Our brain comprises orders of magnitude with a higher number of communicating nodes than recent supercomputers. Furthermore, the dispersion [17] of their communication distances is also by orders of magnitude higher than in supercomputers. A non-spatiotemporal (electronic network) communication model comprising computing nodes, communicating, and cooperating at distances ranging from submicrons to centimeters describes something other than our brain.

### 4.5. Neural Information Content

There is evidence that the neural synapses can be sensitive [33] also to the form of the received spikes. It looks like that, in addition to *membrane’s voltage threshold*, a *synaptic current threshold* also exists (higher synaptic current density or higher local slope of the charge distribution function); notice that the front side of the spike comprises the information again. This conclusion suggests that the spikes may also hide one other bit. A spike with a current slope above the corresponding current threshold value triggers an output spike immediately. That is, it interrupts (and restarts) neural computation. (This mechanism can be an answer to the question “it is unclear how neurons would filter out specific oscillations” [30].) It means that its (primary) information content is if to re-synchronize, independently from the neural computation, the ’bit time’, so any additional information content is neglected. Receiving such spikes can be interpreted as a spike delivering the single payload bit, and whether to restart neuronal computation. The vital information, again, is *when* to restart computation.

### 4.6. Neuronal States as Information

Assuming that the transferred information is about different states in the sender neuron enables us to explain the presence of several bits per spike. There is evidence [53] that the different spiking neurons can stably exist only in a couple of discrete states, including a ground state (GS) and a small number of active state (AS). Provided that we can determine the characteristic Inter-Spike Interval (ISI)s for those different states, one can attempt to interpret that the message comprises the information in which state of the available ones the sender neuron is. Given that in [53] only 4… 5 *stationary* states could be identified, it looks like that 3 bits (in agreement with [4], ch. 6) can describe the transmitted information (in this sense, only those stationary states). Our estimation is that the 3 bits is a population-level value; the neuron level value is probably 1 bit only.

Although this approach provides a perfectly discrete support set, the spikes’ timing information is essentially neglected. We identify some stationary states, do not care at what ISI values the states are located, and omit transient states. Describing the transition from one state to another can carry more information (and need additional bits). It could be a problem to combine these two different contributions.

### 4.7. Spiking Time as Information

The evidence (although based on the classic neural information theory) shows that more than one bit are transferred in a spike. The synchronization signal marks the operand’s arrival time. Still, it can also trigger the restarting of the neural computation (in the periodic operation, it adapts to the local process to a remote base frequency). This part of information belongs to the individual neurons and has the vital attribute of the absolute time of arrival. The rest of the information (the relative timing) belongs to the spike train (the network’s corresponding part), modeled using the ISI. That is, *our reasonable assumption is that an isolated single spike transfers digital information (the synchronization signal, the reset signal, the stationary state index) and the time difference (measured by the internal oscillator used in measuring the ‘bit time’) between spike train members and delivers the rest of the information in quasi-analog form*. The question of how that timing information is coded remains open.

As discussed in Section 3.2, neuron’s spike generation time (its spiking rate) depends sensitively on its inputs from its presynaptic neurons and the neuron’s internal memory. Given that the axon allows a constant ‘conduction velocity’, the spike arrives at the receiver at the same time relative to its previous ‘bit time’ that the presynaptic neuron sent out. That is, the same time difference value (which collects, of course, some noise) is available at the receiver. The relevance of using the time between spikes (ISI) in the experimental works [30,53] strongly supports the relevance of our hypothesis about the role of the temporal description. Whether to measure the time between received spikes (i.e., measuring the spike rate), the receiver neuron uses the ‘bit time’ derived from the previously received time or uses some absolute time that has remained open and is to be researched. We estimate that the time estimation method should adapt to the ‘bit time’.

The present paper suggests a very different approach to the neural information transfer and the information content transferred in a spike: the timing parameters (such as the mean and variance) of the inter-spike interval are the primary carrier of the information. The former approaches (see [4,49] and their cited references), using approximations of limited validity, found that only a few (≈1… 3) bits are transferred in neural spikes. Our approach suggests a direct measurement facility: measuring the distribution of inter-spike periods can prove or deny our assumption.

That is, we propose (following the idea of Hartley [29]) to introduce that the number of distinguishable time values *M* in neural communication is:(4)$$M=1+\frac{F_{Max}}{\Delta F}$$
where the receiver’s rate frequency resolution is ΔF and $F_{Max}$ is the empirical maximum firing rate. In this way, we arrive back to the original definition: *how much information could be resolved after transmission* (i.e., after reconstructing the message)? We tacitly assumed that the neuron could receive and the presynaptic neuron could send spikes with frequency $F_{Max}$; furthermore, the frequency resolution ΔF includes time uncertainties on both sender and receiver sides. We refer to Section 3.4: when measuring ’firing rate’, the convolution of *two* distributions is measured. However, in most cases, the amount of noise probably does not enable performing a more detailed measurement. We can give a rough estimation for the theoretical upper limit for the number of bits that can be transferred in neural communication using ‘firing rate’ coding as 5…6 bits, assuming the frequency of spontaneous firing (≈5 Hz) as a kind of noise and the anatomically possible maximum frequency (≈200 Hz). This estimation is in good accordance with the “just a few bits” maximum, derived experimentally, given that values of parameters of two distributions must be transferred. However, the primary variable is the time of operation, so we must interpret the frequency distribution after performing the reciprocal transformation.

An alternative could be to define an ISI time resolution ΔT (say, the width of the narrowest stationary state) and construct an ISI time histogram from the lowest ISI value to the experienced highest one. In this way, we go back to Hartley’s original definition [29] and create an *N* element ISI histogram. Some stationary states may share their contribution between neighboring cells, and *N* is probably an overestimation (some histogram cells can be empty) of the number of transferred bits
(5)$$N=1+\frac{ISI_{Max}-ISI_{Min}}{\Delta T}$$ If we estimate (from Figure 2 in [53]) ΔT=0.02 s, $ISI_{Max}=0.5$ s, we arrive at N=26, i.e., we can describe the transferred information by 5 (or maybe 6) bits. As discussed in [30], transferring signals with $ISI_{Max}$ repetition rate requires less ATP, so we can presume that, for metabolic efficiency, transferring the (GS) signal needs below-average neural bandwidth and transmission power.

However, we note that although we consider that the message delivers some spiking time-related information that takes us closer to the essence of the true biological information communication, the proposed time histogramming method is arbitrary, and as suggested by Shannon [11], “this is an experimental and not a mathematical fact, and as such must be tested under a wide variety of experimental situations”. In addition, when attempting to crack neural coding, we need to consider, that *if the neuron’s output section applies any type of encoding, the postsynaptic neuron’s input section must use a corresponding decoder*: not only the experimenter, but also the receiver neuron must understand the message. As discussed, *the receiver neuron cannot reconstruct the received message in Shannon’s sense, and even if it could, it would need some mechanism to set the corresponding state communicated in the message*. One more argument to underpin why the “coding metaphor” [16] has limited applicability in neural science is as follows.

## 5. Evidence Underpinning the Suggested Approach

Given that skewed distributions are assumed to have a vital role in the brain’s operation, we show that experimental settings and data evaluation methods can also lead to the effect that otherwise (nearly) normal distributions get (more) skewed. Maybe one shall reformulate the claim “The log-dynamic brain: how skewed distributions affect network operations” [54] as “The dynamic brain: how considering *transition processes* affects network operations” or simply “The short-term learning behind the dynamic brain”. Also, one can experimentally exceed the “power bandwidth” of neuronal communication (see Section 2.6), and in this way imitate symptoms of “infocommunication diseases”.

### 5.1. Neurophysiological Evidence

The learning mechanism discussed in Section 3.5 reveals that short-term and long-term learning are just two sides of the same coin. *The essence of learning is in selective speed-up of communication*. Biology takes advantage of the low speed of information propagation. (An interesting historical parallel that computer EDVAC used delay lines with *msec* processing time for information storage.) Our conclusions seem to be supported by anatomical evidence [42] that “individual anatomical parameters of myelinated axons can be tuned to optimize pathways involved in temporal processing”, and that “the internode length decreases and the node diameter increases progressively towards the presynaptic terminal” and… “these gradations are crucial for precisely timed depolarization”. It was observed [47] that “neuronal activity can rapidly tune axonal diameter” and “activity-regulated myelin formation and remodeling that significantly change axonal conduction properties are most likely to occur over time-scales of days to weeks”.

Biological systems have a partly pre-programmed complex network comprising partly pre-programmed neurons, with well-defined initial synaptic weights (as reflected by their firing rates). As the elegant investigation [48] demonstrated, as an implied side result, the result of learning manifests in that the affected neurons change their firing frequency, using the mechanism described above. Based on our time-aware paradigm [17,27], one could find out directly whether it is long- or short-term learning (whether the change in $T_{Triggering}$ correlates with the change in the corresponding conduction speed or the synaptic strength).

Another direct evidence underpins that the inter-spike time gap comprises *two* components: $T_{Idle}$ and the charging-related sum of the rest of all other times, see Equation (Equation 2). As the right bottom inset of Figure 1 in [1] displays, the standard deviations of the second and third spikes significantly reduce if their *difference* to the time of the first spike is used as a time reference instead of their absolute time. The start signal for time measurement is a transition from negative to a positive velocity of the flight. This signal causes in the individual spikes’ time a “jitter from trial to trial by ∼1 ms”. The jitter time $T_{Idle}$ takes its origin from that this trigger signal is independent of the neural operation’s synchrony. From the point of view of the neuronal operation, the measurement begins at a random time after the refractory period’s end. Subtracting the first spike’s time from the second and third spikes’ times removes this contribution (the “jitter”, presumably a random time and having normal distribution) from the corresponding time values and also reduces those time values’ standard deviation. It is true that “this Self-Information Code is completely intrinsic to neurons themselves, with no need for outside observers to set any reference point such as time zeros of stimulation” [30]. However, the observer can (unintentionally) set that reference point and see its consequences. The diagram lines in Figure 1 demonstrate that the ISI distribution comprises two separable components; furthermore, the standard deviation of $T_{Idle}$ is much larger than that of ISI in the working state. A simple arithmetic jitter correction improves measurement quality by removing the random time $T_{Idle}$ (we estimated the time parameters from [1]). The scattering of $T_{Idle}$ is dominant.

Measurements recording *only* the firing rate attempt to describe these two distributions, having four parameters, with a single distribution altogether. However, we cannot represent the resulting distribution with two parameters; we need to introduce additional parameters, skewness, and kurtosis [30]. Nature requires its degrees of freedom back. The contribution $T_{Idle}$ is mostly characteristic to the given neuron, while $T_{Triggering}$ depends on other neurons’ former state and temporal behavior of their network. The single-distribution model can be a good approximation in a stationary state, that is when the timing relations do not change during the process, but it is definitely not valid in a transient state when either network’s state or neuron’s timing, or both, are changing.

The recent investigation [53] seems to experimentally underpin our claim. In that work, stationary states are called GS and AS, and time (ISI) is used to describe spikes’ distribution. The AS has a higher frequency (i.e., lower ISI), and the neuron approaches the ISI only from one side, from a higher ISI state. Figure 2 shows the ISI distribution for an AS of the neuron. We assume that a neuron stably in AS state can be described by a normal ISI distribution. Similarly, a snapshot of a near-AS state could be described by a similar distribution with a slightly higher ISI mean value, although that state is transient. If we cannot cleanly isolate spikes from ‘AS-state only’ neurons, the measured distribution will not be homogeneous, and the measured distribution will comprise a mixture of distributions from stationary and transient states. The figure shows how this incomplete separation makes the distribution asymmetric and tailed. In other words, mixing ISIs from transient states (processes) to ISIs from stationary states (AS) makes the distribution of ISIs skewed, and the skewness depends on the composition of the mixture. *The transition from GS to AS is essentially a short-term learning process and naturally results in a skewed ISI distribution.*

### 5.2. Mathematical Evidence

With the help of mathematics, biology convincingly underpins that our physics-based model is correct, that is, the primary entity is the operating time instead of the firing rate: how quickly the neuron membrane can reach its threshold potential. In general, “the differences between mean values… are then quantified by statistics based on symmetrical, bell-shaped Gaussian (normal) probability distributions” [54]. However, in the brain “the distribution of both stimulus-evoked and spontaneous activity in cortical neurons obeys a long-tailed, typically lognormal, pattern” [54]. We add that *the pattern belongs to the inverse of the correct probability distribution* (i.e., the activity is measured by ‘firing rate’ instead of ‘computing time’). Although the authors extensively explained why biological processes could obey such a long-tailed pattern, our idea provides a more plausible explanation.

The computing time depends on many factors and comprises contributions $T_{Idle}$ and $T_{Triggering}$. We can presume that, similarly to the overwhelming majority of the many-component biological processes, these times obey the natural *normal distribution*. In active stationary states, contribution $T_{Idle}$ may disappear, and the inter-spike distance obeys the normal (bell-shaped) distribution. However, neurophysiologists usually measure the *firing rate* (that is, the frequency as defined by Equation (Equation 2)) which is the reciprocal of the primary random variable. If we hypothesize that the primary random variable obeys *normal distribution*, we shall use the *reciprocal normal distribution* for fitting the firing rate. In Figure 3, the black diagram lines show the correct distribution for two-parameter combinations (in brackets, the mean and variance values are provided). Unfortunately, the widespread lognormal distribution (red line) has a very similar shape. The two forms show a considerable resemblance, with differences beyond the measurement precision. It is easy to mismatch them, but the idea behind them is entirely different. Notice that if the *firing rate* would obey lognormal distribution, then the *firing time* should obey a power-law distribution. Furthermore the distribution parameters are different; this may lead to different conclusions when calculating mean differences from the distributions.

On the one hand, our reciprocal normal distribution consequently underpins that (under stationary conditions) *computing time obeys normal distribution, and this is why the firing rate obeys the corresponding reciprocal distribution*. On the other hand, we suggest a validating idea. In addition to measuring the firing rate, one shall measure the length of the neuronal computation (that is, the time from the arrival of the first synaptic input to reaching the membrane threshold value). Notice that the frequency also comprises the random variable $T_{Idle}$, the time between the end of the refractory period, and the arrival of the next synaptic input. In experiments similar to the one in [41], one can also control that time. We can agree that “at many physiological and anatomical levels in the brain, the distribution of numerous parameters is in fact strongly skewed with a heavy tail” [54]. However, these parameters may be *derived* from primary parameters with a bell-shaped distribution or, as discussed in Section 5.1, from an inhomogeneous population. Because of this, we do not necessarily agree with “that skewed (typically lognormal) distributions are fundamental to structural and functional brain organization”. However, we agree that ”this insight not only has implications for how we should collect and analyze data”. To understand the brain’s overly complex structure and operation, we need to use primary parameters, a homogeneous neuronal population, and appropriate statistical tools.

### 5.3. Power Bandwidth

As we discussed in Section 2.7, it is not sufficient to satisfy the frequency condition: a power bandwidth can also limit neural communication. In [41], the researchers attempted to estimate the ‘information content’ from a single spike experimentally so that they replaced the presynaptic neuron with a specific artificial current input. However, an artificial input is fed into a natural axon, providing a chance to produce an unconcerted frequency set. The artificial neuron can produce an ‘unnaturally’ high information density (firing frequency). If the experimenter attempts to transmit more information through the connection than the transfer channel has enough energy support for, the transfer hits the ‘energy wall’: part of the ‘information’ cannot get through the channel, and the information seems lost. The neural transfer system is not prepared for such a situation: the ions delivered are stalled (the axon has no mechanism to remove them) and delivered with some ‘recovery delay’, producing ‘unnatural’ symptoms.

The experimenters use a model where the current amplitude represents the information, and there is no memory in the system. The results show that at high firing rates of the artificial input, the *transferred information stays constant*. The axon’s energy supply is not prepared for such high firing rates, and it forms an upper bound for the transmitted information, corresponding to the ‘natural’ possible firing rates. The case is surprisingly similar to the ‘roofline’ experienced in technical computation [31].

Figure 4 in [41] also shows that two close-lying input spikes produce a ‘wider’ spike. That is, when providing another input spike before the relative refractory time ends, the membrane cannot recover between making spikes (the neuron has a memory); therefore, the two spikes are transmitted as one spike (Siamese twins). The effect turns out to be more frequent when a ‘fast’ spiking mode is used. Notice that there is an ‘information loss’ in the number of counts concerning the number of input spikes but not in the area of the spikes (the transmitted charge). ‘Ghost’ spikes (output spikes produced in the absence of input spikes) also appear; when the artificial spike firing is unnaturally high, the transfer channel transmits the remaining excess charge after recovering the power supply cells. The effect of exceeding the channel’s *power bandwidth* causes unsolicited output spikes and other ‘Siamese twins’; overloading its transfer channel forces abnormal behavior of the neuron.

## 6. Discussion

The neuroscience extensively uses the classic information theory to describe neural information transfer and processing. However, the classic information theory was invented for the need of describing electronic communication and many of its features are tacitly derived from that field. In the neural environment, the information and its processing is entirely different. We discussed that, in biology, most of the preconditions needed to apply Shannon’s classic information theory are not met, so the theory is applied far outside of its range of validity. Our conclusions based on the issues concluded from the introduced neuronal model are in complete agreement with the conclusion based on mathematical issues, summarized as “Information theory is abused in neuroscience” [7]. Because of similar fallacies, “the obvious lack of a definitive, credible interpretation [of neural information] makes neuroscience calculations of channel capacity and information transmitted meaningless” [7].

Our idea is that neural information, processing, and communicating means an appropriate adjustment of spiking time. The arrival of a spike delivers a piece of per-spike synchronization information. On the one hand, this bit is technically needed to enable communication at any distance and transfer speed, as discussed in Section 2.8. On the other hand, it is payload information with the content that a spike arrived. Whether the bit representing the spike’s arrival is considered a piece of payload information or service information should be discussed and researched. Spikes’ form may also carry information; for example, to change the spiking phase forcefully [33]. Furthermore, the adjustment of spiking time allows learning, as discussed in Section 3.5.

That is, spikes surely transfer also some digital information, but the *majority of payload information is carried by the spike train and how time is passing between individual spikes*, as claimed in [22]. The reciprocal of that time defines the momentary frequency (the ‘firing rate’) measured in neurophysiology. Our interpretation underpins the findings [2,4,49] that *spike’s precise timing matters* and that *the information transferred in neural communication is primarily temporal*. The careful analysis shows that the firing rate results from the convolution of *two* (partly biased) variables, i.e., it cannot be characterized by *one* single distribution and its temporal pattern cannot directly and uniquely correspond to some “coding”. Its high variability [30] comes from two different sources. Furthermore, our research sheds another light on neural communication’s actual nature: it explains *how* temporal information is delivered.

## Figures and Tables

**Figure 1 entropy-24-01086-f001:**
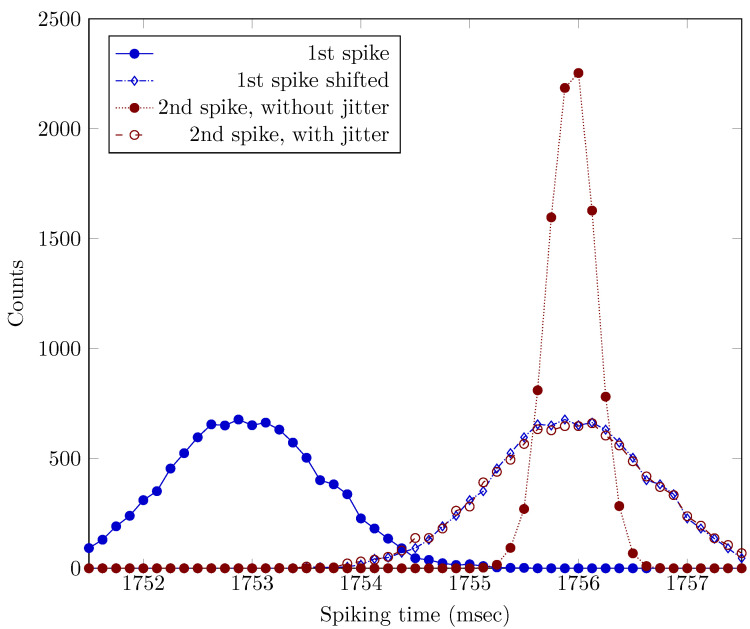
The effect of starting to measure cyclic spiking at a random time vs at a phase-locked time. The random timing contribution $T_{Idle}$ smears the original distribution of the second spike.

**Figure 2 entropy-24-01086-f002:**
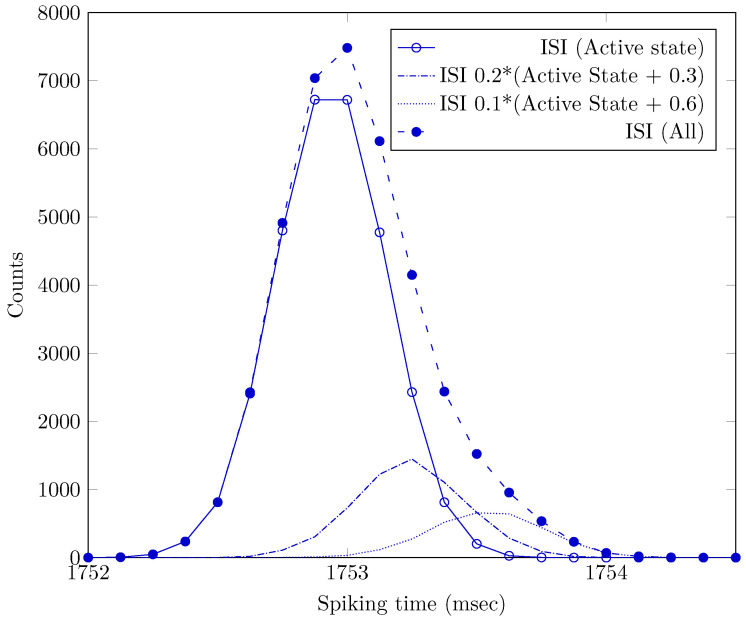
The effect of presence of transient states on the distribution of ISIs from an active state. In the figure, two distributions with intensity of 20% (at AS+0.3) and 10% (at AS+0.6) of the stationary state distribution are contributing to the resulting distribution. Given that active states’ lower ISI value can be approached from the direction of ground states’ higher ISI, the improper separation of AS alone may lead to skewed ISI distribution due to the contribution from spikes in transient states.

**Figure 3 entropy-24-01086-f003:**
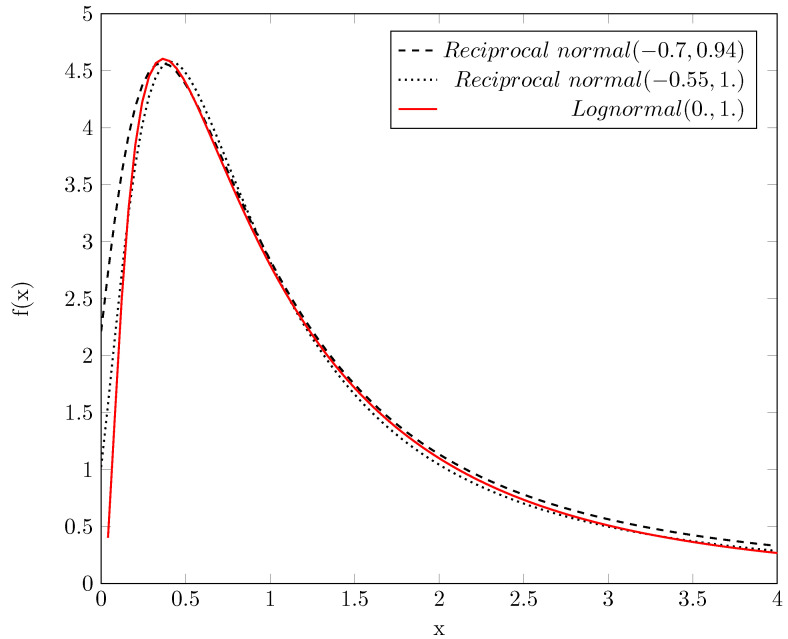
The similarity of the reciprocal normal and lognormal distributions. The shapes’ similarity (although at different parameters) is beyond measurement precision.

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
