# Peer review of "Towards Generalizing the Information Theory for Neural Communication [Author-notes fn1-entropy-24-01086]"

_entropy, 2022, doi:10.3390/e24081086_

Round 1

Reviewer 1 Report

The contribution lists some issues that should be taken into account when applying an information-theoretic perspective to the assessment of neural communication.

 The study is interesting but it should be made more focused and its readability should be improved to attract the attention of the reader.

1)      It is difficult to understand the scope of the contribution. As a review the manuscript does not summarize contributions on the topic and does not classify them according to some criteria. As a regular article it does not provide experimental data and hypotheses that have been tested. As a perspective article it proposes a viewpoint that might be considered not far from other contributions (see e.g. ref #10 and #11) and, consequently, it is unclear the original standpoint. Please clarify.

2)      The contribution should start from the important advancements linked to the exploitation of information-theoretic approaches based on Shannon, conditional and transfer entropies, in decoding dynamical systems interactions including populations of neurons reported in several reviews on the topic (see A.G. Dimitrov et al, J Comput Neurosci, 30, 1-5, 2011; K. Hlavackova-Schindler et al, Phys Rep, 441, 1-46, 2007; A. Porta and L. Faes, Proc IEEE, 104, 283-309, 2016; R. Vicente et al, J Comput Neurosci., 30, 45-67, 2011; E. Pereda et al, Prog Neurobiol, 77, 1-37, 2005) and indicate clearly how faced directly the issues here reported might advance further the field.

3)      It seems to me that part of the controversy about mathematical, technical and biological transfers could be cooled down if the goal is reduced to the study of dynamical characteristics of the process via the assessment of the shape of the distribution and conditional distribution of states as in the pragmatical views reported the above mentioned reviews.

4)      If the purpose of the study is to analyze the suitability of applying information theory to process neural recordings, the authors should list the issues more evidently. I would recommend to the authors to point out the mathematical issue, the technical issue, and, finally, the biological issue consecutively in an assigned subsection. The number of subsections should be as many as the issues and each section should be highlighted with a short title briefly summarizing the issue. This structure should emphasize the issue and the limitations of the theory in practical contexts. In addition, it should attract more powerfully the attention of the reader. This is completely different from the present perspective in which mathematical, technical, and biological issues are listed in separated sessions as a sequence of entries.

5)      It is difficult to understand what is missing whether the issues listed in the present contribution are not faced. The lost information should be made more evident by providing practical examples. In addition, solutions should be proposed on how to tackle the listed issues and how to adapt, if it is the case, information-theoretic metrics to account for the issue.

6)      The contribution should be made more formal to be really useful in practical contexts. For example, the proposed time-aware approach to the information theory is descriptive and generic, while should be rigorously formalized by providing modifications of information domain metrics.

7)      In general, the contribution is rich of information but it is unclear whether all the information provided is necessary to the comprehension of the study and to the final goal. For example, is it really necessary to provide the well-known definition of Shannon entropy for discrete and continuous univariate random variables in (1) and (2)? For example, the need of the entire Sect.2 is unclear, especially because the study exclusively deals with information entropy. The overall contribution needs to be better focused toward the goal and special attention should be devoted to avoid losing the attention of the reader by burdening manuscript with unnecessary details.

Author Response

Comments and Suggestions for Authors

> We provide our replies in-line with reviewer's comments and questions.

The contribution lists some issues that should be taken into account when applying an information-theoretic perspective to the assessment of neural communication.

 The study is interesting but it should be made more focused and its readability should be improved to attract the attention of the reader.

> We have completely restructured the MS  

1)      It is difficult to understand the scope of the contribution. As a review the manuscript does not summarize contributions on the topic and does not classify them according to some criteria. As a regular article it does not provide experimental data and hypotheses that have been tested. As a perspective article it proposes a viewpoint that might be considered not far from other contributions (see e.g. ref #10 and #11) and, consequently, it is unclear the original standpoint. Please clarify.

> We think that the profile of the MS is closest to a research paper.
> There are numerous reviews on applying the classic
> information science to neurons. We introduce a new theoretical
> basis (we show that using time of spiking as information carrier
> is a better approximation than using spike amplitude or firing rate),
> which is alone in its category, so there is no other publication
> to compare to and it is of little use to compare it to publications
> based on the classic theory. Our claim is that the classic information theory 
> is used outside of its range of validity in neuroscience. To underpin
> our claim, we did review the critical aspects referring to reviews and 
> research papers, in this restricted sense making a kind of review.
> It is partly a viewpoint article, too: it introduces the new viewpoint
> that the temporal behavior of neurons explains some not yet 
> understood features, such as how information storing and processing
> occurs at neuronal level, furthermore that we cannot understand
> the behavior of neurons without considering also their network environment
> and vice versa. The another new viewpoint is that we need to consider
> a different quantity as the information carrier, to make using
> information theory to neural systems.
> As a research paper, we made explicitly our hypotheses
> (connected to the commonly used specialized neuronal condenser model),
> that the neuron's operating time (or frequency) comprises two
> components which are contributed by the condenser and its network, respectively.
> We derived a new description mode of neuron's operation
> and we propose to test its consequences experimentally.
> We use published experimental data to test our hypotheses. 

2)      The contribution should start from the important advancements linked to the exploitation of information-theoretic approaches based on Shannon, conditional and transfer entropies, in decoding dynamical systems interactions including populations of neurons reported in several reviews on the topic (see A.G. Dimitrov et al, J Comput Neurosci, 30, 1-5, 2011; K. Hlavackova-Schindler et al, Phys Rep, 441, 1-46, 2007; A. Porta and L. Faes, Proc IEEE, 104, 283-309, 2016; R. Vicente et al, J Comput Neurosci., 30, 45-67, 2011; E. Pereda et al, Prog Neurobiol, 77, 1-37, 2005) and indicate clearly how faced directly the issues here reported might advance further the field.

> We thanks the reviewer for the listed references. We analyzed their results 
> and in different spots, we refer to them.
> Our main point is that we must be careful when exploiting
> information-theoretic approaches based on Shannon, and deriving
> other information-theoretic quantities, because the applicability
> of the theory must be verified case by case.

3)      It seems to me that part of the controversy about mathematical, technical and biological transfers could be cooled down if the goal is reduced to the study of dynamical characteristics of the process via the assessment of the shape of the distribution and conditional distribution of states as in the pragmatical views reported the above mentioned reviews.

> We hope so. The mentioned reviews attempt to describe 
> dynamic characteristics with static notions (see the 
> 'dimensional explosion' in our MS). The 'dynamic brain'
> comprises also processes, not only states. Our suggested 
> description method hopefully can help in resolving the controversy.

4)      If the purpose of the study is to analyze the suitability of applying information theory to process neural recordings, the authors should list the issues more evidently. I would recommend to the authors to point out the mathematical issue, the technical issue, and, finally, the biological issue consecutively in an assigned subsection. The number of subsections should be as many as the issues and each section should be highlighted with a short title briefly summarizing the issue. This structure should emphasize the issue and the limitations of the theory in practical contexts. In addition, it should attract more powerfully the attention of the reader. This is completely different from the present perspective in which mathematical, technical, and biological issues are listed in separated sessions as a sequence of entries.

> It is correct. Actually, the different fields and the issues are
> like a two-dimensional array. We made that array linear in the wrong way.
> In the new way, the issues and differences are more comprehensible.

5)      It is difficult to understand what is missing whether the issues listed in the present contribution are not faced. The lost information should be made more evident by providing practical examples. In addition, solutions should be proposed on how to tackle the listed issues and how to adapt, if it is the case, information-theoretic metrics to account for the issue.

> "how to adapt, if it is the case". Unfortunately, it is not the case.
> As it is now detailed in the MS, the messages comprise !two! components:
> a temporal one and the signal. In biology, the temporal component
> is decisive. Adapting a two-component message to a one-component one,
> is not possible. This is what in electronics the "clock distribution tree"
> makes. The price is the tremendous heat production in today's processors.
> In the mathematical theory, there is no time component. Unfortunately,
> the mathematical formulas always provide a result. This is what
> Shannon warned about:
> "is not a trivial matter of translating words to a new domain".

6)      The contribution should be made more formal to be really useful in practical contexts. For example, the proposed time-aware approach to the information theory is descriptive and generic, while should be rigorously formalized by providing modifications of information domain metrics.

> We hope the contribution is more formal now.
> "to be really useful in practical contexts", our suggested hypothetic
> information carrier should be thoroughly tested. As Shannon said:
> "this is an experimental and not a mathematical fact, and as such 
> must be tested under a wide variety of experimental situations",
> although the paper
> attempts to formulate it in math notions.
> However, the question remains open, if the nature uses same notions.

7)      In general, the contribution is rich of information but it is unclear whether all the information provided is necessary to the comprehension of the study and to the final goal. For example, is it really necessary to provide the well-known definition of Shannon entropy for discrete and continuous univariate random variables in (1) and (2)? For example, the need of the entire Sect.2 is unclear, especially because the study exclusively deals with information entropy. The overall contribution needs to be better focused toward the goal and special attention should be devoted to avoid losing the attention of the reader by burdening manuscript with unnecessary details.

> Correct.
> The well-known definitions replaced with reference to a good textbook. 
> We do hope we could make the MS better focused.

Reviewer 2 Report

This study proposes to set the use of information theory in neuroscience on a more rigorous foundation, including by reconceptualizing the neuronal firing rate in terms of its underlying time-resolved constituents. This is potentially an interesting and original investigation, and could provide an important alternative viewpoint on information theory in neuroscience. Having said this, as a practicing neuroscientist, I don't think in it's current presentation the manuscript will reach and convince neuroscientists of its merits, which is presumably its overall intended aim. I therefore strongly suggest that the authors revise the writing and presentation to make the presentation more balanced and neuroscientifically relevant.

Specifically, the paper comes across to state that (1) faulty application of information theory underlies the major failures of neuroscience, that (2) such faulty applications seem largely unappreciated in the field, and that (3) the present paper considerably clarifies these faulty applications.

With regard to (1) the paper cites for example the failure of recent whole-brain simulations, without considering that many other absent parts of these simulations (including their grounding to cognition or behavior) that have caused these failures, independently of our knowledge (or lack thereof) of information theory. The paper also cites the seemingly contradictory claims about rate- and time-"coding", without considering that both types of "code" can play important and complementary roles in neural activity (something many neuroscientists arguably understand quite well). Elsewhere, it talks that models in neuroscience do not take into account discussion of bandwidth limitations, while citing the literature from a neuroscience field that does not focus on information theory or neuronal communication. More generally, much of neuroscience does not rest on presumed assumptions of information theory, and I urge the authors to make the discussion of this literature more accurate and targeted in scope, and correspondingly more balanced.

With regard to (2) the paper omits well-known literature that has already critiqued the use of information theory in neuroscience. The authors do not cite for example, Brette (2019) "Is coding a relevant metaphor for the brain?", an expansive article that goes indeed beyond the present discussion by arguing that the notion of coding in neuroscience should be dismantled altogether. The Brette article comes with a multitude of responses and replies, is highly relevant to the present paper, and indeed resolves some of the seeming contradictions discussed in the present paper by showing how to reconcile aspects of "rate code" and "time code". It, and other associated literature in neuroscience, should clearly be referenced in the context of the present work.

With regard to (3) the authors neglect to mention the validity of their tacit assumptions, and instead use fairly strong language that other studies "prove our claim", or "biology provides evidence that our model is correct", or "we can presume that these times obey the normal distribution". Hidden within these statements are questions of validity, alternative explanations that are not stated, and more generally the ill-defined notion of proof in biology (we can at best tentatively confirm models, but never prove them). A thorough discussion of these underlying assumptions and limitations is in order. 

As a general addition to these points, I think the presentation would considerably benefit from a more structured organization that would clearly summarize:

* The main problem the authors are trying to resolve early on in the study ("information theory is abused in neuroscience" is too vague and nonspecific).

* A clear definition of the model, and it associated assumptions. It would help to have all the assumptions summarized in a single location, perhaps even a table or box. This would allow the readers to readily appreciate the limitations of these assumptions.

* A clear discussion of the results, including all the relevant methodology that was produced to make the Figures. The present discussions of the figures is very sparse, and it was unclear in places how the results were actually derived.

* Results for existing competing models that the authors propose to overturn, and objective comparisons to these models, including with statistics. For example, Figure 3 shows the relationship between reciprocal normal and lognormal distributions, but this is misleading because only one “tail” of the reciprocal normal is illustrated.

* Limitations of the present formalism, including validity of the assumptions.

Overall, I hope that these suggestions will make it easier for neuroscientists to understand the authors overall intent, and in this way appreciate the overall potential significance of their contribution.

Author Response

Comments and Suggestions for Authors

> We write our replies in-line with reviewer's comments and questions.

This study proposes to set the use of information theory in neuroscience on a more rigorous foundation, including by reconceptualizing the neuronal firing rate in terms of its underlying time-resolved constituents. This is potentially an interesting and original investigation, and could provide an important alternative viewpoint on information theory in neuroscience. Having said this, as a practicing neuroscientist, I don't think in it's current presentation the manuscript will reach and convince neuroscientists of its merits, which is presumably its overall intended aim. I therefore strongly suggest that the authors revise the writing and presentation to make the presentation more balanced and neuroscientifically relevant.

> Well, it is the usual problem of multidisciplinary research.
> The aim is really to introduce a new merit of information into neuroscience,
> but the first step is to establish it mathematically. The second step
> really should be to achieved with "neuroscientifically relevant" descriptions.
> However, our MS is a theoretical work, with no own measured data.
> It is very hard to find published experiment, with enough published conditions.
> The experiments are designed having a different hypothese in mind.
> As the "Coding metaphor" paper formulates:
> "neural code depends on experimental details that are not carried
> by the coding variable" and "spikes are timed actions that mediate
> coupling in a distributed dynamical system".
> Now, having the idea, "a practicing neuroscientist" has an excellent
> possibility to confirm or deny our conclusions.

Specifically, the paper comes across to state that (1) faulty application of information theory underlies the major failures of neuroscience, that (2) such faulty applications seem largely unappreciated in the field, and that (3) the present paper considerably clarifies these faulty applications.

> The really strange with the "faulty application of information theory"
> is that, in one set of conditions, the classic information theory
> may result accidentally some correct results (see the special cases
> of the neurons operating time), with a slightly different other set
> again some seemingly correct results, but the conclusion in the 
> two cases may be different. 

With regard to (1) the paper cites for example the failure of recent whole-brain simulations, without considering that many other absent parts of these simulations (including their grounding to cognition or behavior) that have caused these failures, independently of our knowledge (or lack thereof) of information theory.
> Sorry for the incomplete idea. Now we removed that sentence form the MS.
> It just makes the MS less focused.
> Concerning the idea, in somewhat more complete form: 
> Please see van Albada et al, Frontiers in Neuroscience, 12(2018)291 
> "This means that any studies on processes like plasticity,
> learning, and development exhibited over hours and days of biological time
> are outside our reach."
> The technically low performance does not enable even to begin
> to work with cognition or behavior simulations.  
> The technically low performance has physical reasons, see
> https://arxiv.org/abs/2001.01266
> Correct:
> "independently of our knowledge (or lack thereof) of information theory",

> the needed computing performance does not enable to begin that level of simulation. 

The paper also cites the seemingly contradictory claims about rate- and time-"coding", without considering that both types of "code" can play important and complementary roles in neural activity (something many neuroscientists arguably understand quite well).
> We are pretty sure that "neuroscientists arguably understand quite well
> neural activity". The reviewer also uses quotations around "coding".
> In communication sense, and without quotation marks, coding needs
> (see the discussion in the revised MS) a coder and a decoder,
> in the output section and the input section of the neuron, respectively.
> The "coding" is not coding in the sense of information theory.
> It is just a signal pattern shown to the observers, which is connected to
> the experienced neural activity through their human intelligence.

Elsewhere, it talks that models in neuroscience do not take into account discussion of bandwidth limitations, while citing the literature from a neuroscience field that does not focus on information theory or neuronal communication. More generally, much of neuroscience does not rest on presumed assumptions of information theory, and I urge the authors to make the discussion of this literature more accurate and targeted in scope, and correspondingly more balanced.

> Indeed, we do not know about that the bandwidth for neuroscience
> is interpreted as we do, with considering conduction velocity
> and spike length. Under healthy natural conditions, there is no
> question about the bandwidth.
> In Shannon's work "the number of allowed signals of duration T" mentioned.
> The theory was inspired by electronic communication;
> so the EM wave propagation speed was assumed.
> Obviously wrong for neural communication.

With regard to (2) the paper omits well-known literature that has already critiqued the use of information theory in neuroscience. The authors do not cite for example, Brette (2019) "Is coding a relevant metaphor for the brain?", an expansive article that goes indeed beyond the present discussion by arguing that the notion of coding in neuroscience should be dismantled altogether. The Brette article comes with a multitude of responses and replies, is highly relevant to the present paper, and indeed resolves some of the seeming contradictions discussed in the present paper by showing how to reconcile aspects of "rate code" and "time code". It, and other associated literature in neuroscience, should clearly be referenced in the context of the present work.

> Thanks for the hint. It is really a great paper, but it is "negative":
> it shows why, in general, the coding metaphor is wrong. Its conclusions
> are correct. Our paper is "positive" and goes behind that idea:
> we discuss from the 'first principles' how the firing rate behaves.
> We agree, that in general, we cannot conclude a "code" from the firing rate. 
> The complex interaction between the neuron and its environment
> (the many inputs and the spatiotemporal behavior) does not enable
> to provide general results. However, for special cases we can
> produce theoretical behavior, and can compare with experimental one,
> provided that enough experimental details are known.

With regard to (3) the authors neglect to mention the validity of their tacit assumptions, and instead use fairly strong language that other studies "prove our claim", or "biology provides evidence that our model is correct", or "we can presume that these times obey the normal distribution". Hidden within these statements are questions of validity, alternative explanations that are not stated, and more generally the ill-defined notion of proof in biology (we can at best tentatively confirm models, but never prove them). A thorough discussion of these underlying assumptions and limitations is in order. 

> Sorry for the strong language; fixed.

As a general addition to these points, I think the presentation would considerably benefit from a more structured organization that would clearly summarize:

* The main problem the authors are trying to resolve early on in the study ("information theory is abused in neuroscience" is too vague and nonspecific).
> Sorry, it is cited as a title of a paper from a renowned and refereed journal with information theory profile. Fixed and moved.
> Unfortunately, Shannon himself protested against using his ideas
> in the way as (among others) neuroscientists do:
> "is not a trivial matter of translating words to a new domain"
> The aim of our paper is to provide that nontrivial translation.

* A clear definition of the model, and it associated assumptions. It would help to have all the assumptions summarized in a single location, perhaps even a table or box. This would allow the readers to readily appreciate the limitations of these assumptions.

> Correct. Now the MS discusses the specialized neural condenser
> in sufficient detail.

* A clear discussion of the results, including all the relevant methodology that was produced to make the Figures. The present discussions of the figures is very sparse, and it was unclear in places how the results were actually derived.

> Fixed

* Results for existing competing models that the authors propose to overturn, and objective comparisons to these models, including with statistics. For example, Figure 3 shows the relationship between reciprocal normal and lognormal distributions, but this is misleading because only one “tail” of the reciprocal normal is illustrated.

> The measured data are interpreted only in the positive domain.

* Limitations of the present formalism, including validity of the assumptions.

> We consider that our assumptions are valid as long as the 
> neural condenser model is valid

Overall, I hope that these suggestions will make it easier for neuroscientists to understand the authors overall intent, and in this way appreciate the overall potential significance of their contribution.

> Many thanks for recognizing the potential significance of our paper
> and the useful suggestions.  Also for some (well deserved) critique.

Round 2

Reviewer 1 Report

The manuscript is improved but, being its nature closer to a research paper as stated by the authors, it would require simulations to support some statements, that otherwise remain generic. For example, the notion that the introduction of time of spiking as information carrier is better than an approximation using solely spike amplitude or firing rate should be proven at least in the context of simulation. Its apparent triviality does not assure its soundness and, even more importantly, does not help the reader to implement the new framework/conceptualization. In addition those simulations might allow to distinguish the proposed framework from traditional applications of information metrics in spike analysis.

Author Response

>  Thanks for the suggestion and evaluating the corrections we made.
> The answers are provided below inline with reviewer's opinion.

The manuscript is improved but, being its nature closer to a research paper as stated by the authors, it would require simulations to support some statements, that otherwise remain generic. 
For example, the notion that the introduction of time of spiking as information carrier is better than an approximation using solely spike amplitude or firing rate should be proven at least in the context of simulation.

> The difficulty with this requirement is that the idea itself is brand new
> (it is closely related to the option of using spatiotemporal distance
> instead of some absolute time measured at some unknown position)
> We could derive the two different merits for the information carrier if data 
> for deriving those merits were available and the communication models used in both cases
> were identical. Even if we could carry out both calculations, the problem
> persists how to compare the result of a good approach to a bad one.
> (actually, the wrong approach calculates the "neural information" from the
> non-payload synchronization signal, while the good one uses the payload signal timing).
> Some math-only procedure could be followed, say showing what amount of information
> could be derived from a signal with different temporal information, but 
> given that the two conduction speed values are million times different 
> (using intermediate values has no biological counterpart) would not be really
> relevant: we compare the technical and biological cases, which use different
> approximations. The biological approximation provides wrong value if applied to
> technical communication and vice versa. 

> Per Reviewer's request, an extra section is added, discussing the different approximations used in biology and electronics.

Its apparent triviality does not assure its soundness and, even more importantly, does not help the reader to implement the new framework/conceptualization.

> It is a pleasure for us that the reviewer considers our proposal a triviality 
> (although according to our best knowledge, we are the first suggesting that idea).
> Given that such dedicated measurements (to our best knowledge) do not exist yet,
> at the best, we could provide some arbitrary assumed parameters and distributions.
> The idea is to measure T_Charging and T_Refractory in advance and
> to derive the rest of time contributions by
> defining a "begin computing" signal (the arrival of the first input spike
> and the end of the absolute refractory period, respectively)
> and the "end computing" signal (when the spike preparation begins) and record the respective
> time distributions. One can compare if the mathematical removal of T_Idle and setting
> the "begin computing" experimentally result in the same distribution for T_Idle.
> One could correlate the input currents gated by signals "begin computing"
> and "end computing", to show that the integrated sum is proportional to 
> membrane's voltage. Furthermore, one can see if neuron's memory really depends
> on the "past" as seen by the neuron: if it raises proportionally as the relative 
> timing of the input spikes changes; if it remains constant if
> the net state is kept constant. Similarly, it should be demonstrated experimentally,
> that the input spikes are truncated when the membrane's threshold reached.
> Concerning learning, we have non-molecular simulations, but again 
> one should experimentally measure how the concentration gradient in 
> the synaptic gap changes (both at active and inactive synapses).
> Here the simulation cannot give advises to the nature.
> Here experiment should have biological relevance, math simulation could
> only prove the internal consistency of the model and its implementation.
> Our proposal, as formulated, needs to be confirmed or denied experimentally,
> as Shannon's requirement said "this is an experimental and not a mathematical fact,
> and as such must be tested under a wide variety of experimental situations".
> Our hypothesis about how experimental data shall be evaluated should be
> confirmed/denied by experiment. Making numerical histograms would not 
> validate our hypothesis. Experimentalists should check how much our parameters
> for resolution, minimum and maximum values are valid, and how much the
> experimental histograms are informative.
> However, we agree with the reviewer, that after that the measurable conclusions
> of the model validate our assumptions, the simulation using the
> already validated model will be an indispensable tool to complement 
> neurophysiological measurement.

In addition those simulations might allow to distinguish the proposed framework from traditional applications of information metrics in spike analysis.
> Such simulations are possible at several levels and depts.
> We provided two simple simulations in the paper.
> The difficulty of simulating fly's triggering effect, only the distribution parameters
> were provided in the paper, and the authors did not reply to our request for data. So, we had to 
> proceed with the available data, which actually means simulating the distributions 
> for the first two spikes. With all available published data we have the difficulty that 
> they do not have the details needed for us approach. At the time of measuring the data,
> it was simply not a relevant point for the respective authors to consider such aspects.

> As can be concluded from the paper, one cannot understand the operation of a neuron
> without its network, and vice versa. Correspondingly, we cannot simulate them with the
> presently available simulators. For that goal, a new simulator must be developed
> from the "first principles". It must be event driven and imitate the temporal behavior
> of both the neuron and its network. We estimated the development cost of such a simulator
> as several dozens man-months. In the frame of the present grant, we did the first steps
> towards  implementing it (in SystemC+Qt) and we are at about 60% of the road,
> expecting to finish it (to its first usable state) in a half year.

>

Round 3

Reviewer 1 Report

The authors replied satisfactorily to my final issue and took into account the suggestions given.